# Will Joining Cooperative Promote Farmers to Replace Chemical Fertilizers with Organic Fertilizers?

**DOI:** 10.3390/ijerph192416647

**Published:** 2022-12-11

**Authors:** Guangcheng Wei, Xiangzhi Kong, Yumeng Wang

**Affiliations:** School of Agricultural Economics and Rural Development, Renmin University of China, Beijing 100872, China

**Keywords:** cooperatives, smallholder farmers, chemical fertilizers, organic fertilizers, endogenous conversion model

## Abstract

This study examines whether cooperatives can assist more than 200 million farmers in China, who are facing tightening resources and environmental constraints, in adopting green farming practices. A framework for counterfactual analysis was established to quantify the impact of farmers joining cooperatives on the reduction in chemical fertilizer consumption and the use of organic fertilizers. The study’s conclusions are based on data from 712 farmers in four counties in Shandong and Henan provinces. Joining a cooperative can have a positive impact on farmers’ selection of environmentally friendly production methods, which increases the likelihood of farmers reducing chemical fertilizer application by 35.6% and organic fertilizer application by 22.0%. It can also reduce the cost of chemical fertilizer application by an average of $209.2/ha. The extent to which smallholder farmers use chemical and organic fertilizers after joining cooperatives depends on the size of their farming operations and their perception of green production.

## 1. Introduction

As global food security remains a pressing concern, many nations are using more fertilizer to increase food production. Global agriculture consequently faces unprecedented dangers and problems [1,2]. Farmers are increasing their input of fertilizers, insecticides, and other energy sources to alleviate the demand crisis [3,4]. The overuse of chemical fertilizers, particularly nitrogen fertilizers, has caused significant environmental damage, as has the excessive use of chemical compounds [5,6]. Specifically, the overuse of nitrogen fertilizers threatens air, water, and soil quality, and biodiversity by causing the eutrophication of aquatic bodies [7]. Consequently, the tension between providing food security and preserving ecosystems is gaining prominence in academic research [8,9]. Although China’s grain production capacity is rising significantly, the excessive or unnecessary use of fertilizer is also of great concern [10]. In 2015, agricultural pollution in China surpassed industrial pollution as the primary source of water pollution [11], which has become a significant concern in the context of food security [12]. Moreover, the dosage of chemical fertilizers used in China is higher than the international environmental safety limit, which is more than twice that of Japan and 6–7 times that of the United States and the European Union [13]. This increase in environmental pollution not only deprives farmers of economic rewards and decreases the safety of agricultural products but also places a substantial strain on the natural environment [5]. Decreasing the use of chemical fertilizers or opting for the use of organic fertilizers as a green agricultural production strategy might mitigate agricultural pollution [10,14,15,16]. The Chinese government has implemented a variety of measures to support a decrease in chemical fertilizer usage, increase sustainable agricultural development, and encourage the transition of agricultural output to green production [17].

Small farmers are more likely to apply more chemical fertilizer to increase crop yield [18,19,20]. First, the production methods of smallholder farmers are plagued by closed operations, low scale efficiency, insufficient worker productivity, and uncompetitive agricultural products [21]. There are also insufficient incentives for smallholder farmers to acquire new information and technology, which results in the relatively limited use of agricultural chemicals [19,22].

Second, part-time farming is associated with excessive fertilizer application. With the advent of urbanization in China, an increasing number of farmers leave their farms to find employment and are less willing to acquire new skills and adopt new technologies that could enhance their farming operations [14,22]. In the case of farmers with off-farm income, the benefits from new technologies are significantly less than their off-farm income; therefore, they will choose to increase fertilizer use, reduce agricultural labor inputs, and may even overuse fertilizers to increase yields [17,19,20,23].

Third, smallholder farmers often have lower levels of education. Their lack of knowledge on pesticide toxicity causes them to apply far more pesticides than professional farmers [24,25,26]. Smallholder farmers also do not make sufficient long-term investments in agriculture. The relatively low income of smallholder farmers and the lack of access to suitable transportation are not favorable to the initial investment in a variety of irrigation and tillage equipment or the promotion and usage of technologies such as organic fertilizers [14,27]. Owing to the high cost of fixed asset inputs, smallholder farmers are more likely to use chemical fertilizers to boost crop yields [22,28]. Farmers are also unwilling to embrace green production technology independently [29]. Despite the introduction of soil tests and the elimination of fertilizer subsidies in China, smallholder farmers are still inclined to use fertilizers in high quantities when compared to professional farmers [20].

Cooperatives have become a crucial organizational mechanism for small-scale farmers, as they provide a solid foundation for farmers’ agricultural production techniques, expand their options to implement green technologies, and foster sustainable agricultural development [20]. Through the division of labor, cooperatives enable farmers to concentrate on productive activities, withstand market risks, and receive returns to scale, which is crucial to promoting cleaner production methods [29,30]. Cooperatives are spontaneously formed economic organizations based on geographical, kinship, and affinity ties [31]. Cooperative farmers also have greater access to innovative agricultural technologies [29,32,33]. Cooperatives can encourage farmers to make use of advanced technologies and share experiences, such as using superior seed varieties [34,35], water-saving irrigation systems, the scientific use of fertilizers [36], and adopting environmentally friendly technologies [29]. However, some academics have divergent perspectives. The generally small size of smallholdings, their loose structure, irregular governance, and weak competition within cooperatives [37,38] have given smallholder farmers’ less access to services, which has further constrained their profit margins [39]. However, there are still some well-run cooperatives that can play a strong driving role for small farmers [40,41,42]. For farmers, reducing the use of common fertilizers and increasing the use of organic fertilizers are one of new kinds of agricultural production technologies. However, there are few studies on farmers increasing the use of organic fertilizers and the substitution of chemical fertilizers to organic fertilizers by joining cooperatives. Therefore, this study focuses on whether Chinese cooperatives can encourage smallholder farmers to reduce their use of chemical fertilizers and adopt organic fertilizers.

## 2. Theoretical Mechanism

### 2.1. Scale Effects

Cooperative societies help to increase the scale of farmers’ operations by increasing the selling price of their agricultural products and helping them to adopt new agricultural technologies by purchasing agricultural production materials for farmers in a coordinated manner [43]. This improves farmers’ returns and increases the scale of their operations [36,44]. Moreover, large-scale operations can contribute to environmental sustainability [22], minimize fertilizer consumption [25], and provide substantial ecological benefits [20]. Large-scale farms can therefore both boost yields and reduce fertilizer use [14,22,25].

Unlike smallholder farmers, large-scale farmers plant crops to pursue greater economic benefits and to enhance their dependence on the land, and hence invest in green production with long-term benefits for the land [20]. The costs of learning and implementing green agriculture methods vary among farmers with large holdings [19]. Large-scale farmers can reduce transaction costs, so they can obtain new agricultural materials at a lower cost [45]. Compared with small-scale farmers, large-scale farming operations utilize larger areas of land and therefore have a lower average costs per hectare for introducing green technologies [19]. In contrast, small-scale farmers face the problem of land fragmentation and do not acquire the benefits of economies of scale, which hinders the adoption of green production practices [46]. Large-scale agricultural operators can meet the scale criteria for promoting green production technologies [27] and often have superior infrastructure, which makes it easier to adopt new technologies [19].

### 2.2. Learning Effects

By joining cooperatives, farmers can gain access to new technologies and information, which reduces the costs of learning new technologies [32], promotes specialization and cooperation in production, decreases the frequency of common pesticide applications [47], and reduces agricultural pollution [29]. Small-scale farmers find it difficult to introduce expensive new technology and are therefore less unwilling to adopt green production systems independently [10]. Cooperatives provide technical guidance and help with the acquisition and adoption of green industrial technologies [48]. Farmers also develop social networks through cooperatives, which allows for the transfer of a diverse range of information and communication skills [31]. Observation and imitation are the primary means by which farmers acquire new information and technologies [49]. The transmission of scientific practices through the social networks of cooperatives makes farmers more aware of the risks and losses of excessive fertilizer application [29], which can assist them in reducing their fertilizer usage [50], and increase their use of organic fertilizer [51].

### 2.3. Regulatory Effects

The uniform marketing of agricultural products is an essential role of a cooperative organization [52]. To incentivize members to produce green agricultural products, cooperative organizations implement a tiered purchasing structure [53]. This system of incentives and limits encourages farmers to adopt more environmentally friendly production methods [54]. Chinese agricultural products suffer from quality and safety issues related to the incorrect use of chemical fertilizers, making it more difficult to meet the market demand for quality agricultural products [55]. This market imbalance between supply and demand is an incentive for cooperatives to enhance the quality of their members’ agricultural products to meet the demand for sustainable products [56]. Therefore, a cooperative society establishes quality standards for agricultural products and requires its members to provide agricultural products that meet the quality requirements [57]. Additionally, a cooperative society inspects the quality of the agricultural products sold through cooperatives and acquires high-quality products from farmer [58]. To satisfy market demand, maintain brand image, and guarantee product quality, the cooperative society monitors its members’ fertilizer application techniques and encourages them to adopt green agricultural practices [59]. Therefore, the whole theoretical framework is illustrated in Figure 1.

## 3. Data Sources and Model Setting

### 3.1. Data Sources

The data used in this empirical study was obtained from a field survey of rural households in Shandong and Henan provinces that was conducted from April to July 2020. Four prefecture-level counties in Shandong and Henan Provinces (Cao, Chengwu, Tangyin, and Hua) were selected as survey sites because of their representative grain planting scale and adjacent latitude (Figure 2). Shandong and Henan are two key grain provinces in the Central Plain of China, which indicates our study site is a good representation. Three towns were selected in each county, and three villages were selected in each town. Specifically, we chose Pulianji, Qingguji and Taoyuanji in Cao County; Wenshangji, Baifutu and Datianji in Chengwu County; Han Zhuang, Baiying and Guxian in Tangyin county; Daokou, Baidaokou and Liugu in Hua county. For each village, we randomly selected 20–25 farmers for one-on-one interviews. Stratified sampling was used to select samples based on economic development, geographic dispersion, and agricultural characteristics. The questionnaire was processed through one-to-one interviews between well-trained investigators and anonymous interviewees. Interviewees filled out questionnaires according to the facts. The interview places were all arranged in local closed offices that were conducive to the interviewees speaking freely. A total of 763 questionnaires were completed, after removing questionnaires with missing information and logical errors, 712 valid samples were utilized, with an effective rate of 93.32%. To eliminate discrepancies regarding the usage of fertilizers caused by different planting types, only wheat farmers were selected to participate in this study.

### 3.2. Model Setting

#### 3.2.1. Endogenous Conversion Model

When deciding whether to join a cooperative, it is impossible to determine whether the same farmer uses fertilizer at the same time. Therefore, it is impossible to directly evaluate the influence of cooperative membership on the fertilizer application behavior of farmers. In addition, farmers’ participation in the cooperative is not determined by chance, but rather by the optimal decision they make under numerous limitations, which is defined by self-selection. Non-randomized controlled trials are incapable of producing counterfactual outcomes [60]. Prior research mostly employed propensity score matching (PSM) to address the issue of selective bias [61]. This strategy cannot resolve endogenous issues brought on by unobservable variables [10,29]. Considering the selection bias caused by observable and unobservable factors, this study refers to the research of LokShin and Sajaia [62], Abdulai and Huffman [61] and Ma et al. [10,29], and empirically studies the effect of joining cooperatives on farmers’ fertilizer application behavior by using endogenous switching Probit model and an endogenous switching model.

This study developed an endogenous switching model, an endogenous switching probit model, and a counterfactual analysis framework based on regression results to estimate the treatment effect of joining a cooperative on farmers’ likelihood of opting for green production methods.

Whether a farmer chooses to join a cooperative is taken as the treatment variable $S_{i}$; its value is 1 if a farmer accepts the treatment and 0 otherwise. Whether a farmer accepts treatment can be expressed by Equation (1).
(1)$$S_{i}^{*}=Z_{i}\alpha \;+\;\mu_{i},\;S_{i}=\left\{\begin{array}{c} 1,\;S_{i}^{*}>0\\ 0,\;S_{i}^{*}\leq 0 \end{array}\right.$$
where $S_{i}^{*}$ denotes the unobservable latent variable of a farmer’s choice to join the cooperative; $Z_{i}$ denotes the variable that affects a farmer’s choice to join the cooperative; α denotes the coefficient to be estimated, $\mu_{i}$ is the random disturbance term, and $S_{i}$ represents the observed decision outcome of whether a farmer joined the cooperative. $S_{i}$ = 1 indicates the joining of the cooperative; if a farmer does not join the cooperative, $S_{i}$ = 0.

Following this, the outcome equation of farmers’ adoption of green agricultural practices in various states is defined in Equation (2).
(2)$$Y_{i}\;=\;\left\{\begin{array}{c} y_{1i},\;S_{i}=1,\;y_{1i}=I,\;y^{*}{}_{1i}=X_{1i}\beta_{1}+\varepsilon_{1i}\\ y_{0i},\;S_{i}=0,\;y_{0i}=I,\;y^{*}{}_{0i}=X_{0i}\beta_{0}+\varepsilon_{0i} \end{array}\right.$$
where $y^{*}{}_{1i}$ and $y^{*}{}_{0i}$ denote the latent variables of a farmer’s choice of green production methods when joining a cooperative and when not joining a cooperative, which determine the observed binary green production method state variables $y_{1i}$ and $y_{0i}$, respectively. $X_{1i}$ and $X_{0i}$ are covariates that influence a farmer’s choice of using environmentally friendly production methods. $\beta_{1}$ and $\beta_{0}$ are the coefficients to be estimated. $\varepsilon_{1i}$ and $\varepsilon_{0i}$ are random disturbance terms with a mean 0, which assumes they all follow a normal distribution. The correlation coefficient between $\varepsilon_{1i}$ and $\mu_{i}$ is $\rho_{1}$, and the correlation coefficient between $\varepsilon_{0i}$ and $\mu_{i}$ is $\rho_{0}$.

#### 3.2.2. Estimation of Treatment Effects

Considering the correlation between the error components of the selection equation and the outcome equation, this study adopted the log-likelihood function developed by Lokshin and Newson [62] and employed the method of the great likelihood to generate consistent estimates. In this instance, the treatment effect of sample inclusion in the treatment group ($S_{i}$ = 1) on the probability of selecting green production practices in cooperatives is shown in Equation (3).
(3)$$\mathrm{TT}=\frac{\emptyset (X_{1}{\hat{\beta }}_{1},\;Z\hat{\alpha },\;{\hat{\rho }}_{1})\;-\;\emptyset (X_{0}{\hat{\beta }}_{0},\;-Z\hat{\alpha },\;{\hat{\rho }}_{0})}{\emptyset \left(Z\hat{\alpha }\right)}$$

The treatment effect of sample inclusion in the control group ($S_{i}$ = 0) on the probability of selecting the green production method is shown in Equation (4).
(4)$$\mathrm{TU}=\frac{\emptyset (-X_{1}{\hat{\beta }}_{1},\;Z\hat{\alpha },\;{\hat{\rho }}_{1})\;-\;\emptyset (X_{0}{\hat{\beta }}_{0},\;-Z\hat{\alpha },\;{\hat{\rho }}_{0})}{\emptyset \left(-Z\hat{\alpha }\right)}$$

The treatment effect of all the samples joining the cooperative on their likelihood of using green manufacturing methods is shown in Equation (5).
(5)$$\mathrm{TE}(x)=\emptyset (X_{1}\beta_{1})\;-\;\emptyset (X_{0}\beta_{0})$$

### 3.3. Variable Setting and Descriptive Statistics

#### 3.3.1. Core Dependent Variables

This study established two dependent variables. The first dependent variable of whether farmers will use chemical fertilizers is based on the research of Emmanuel et al. [63]. Furthermore, it also examines whether farmers’ participation in cooperatives helps to minimize chemical fertilizer use. The amount of fertilizer purchased per hectare is used as an indicator of the extent to which farmers minimize fertilizer application based on the research of Ma et al. [29]. This study examined farmers’ use of organic fertilizer as a crucial dependent variable to determine whether organic fertilizer application was a green production practice based on the research of Ma et al. [10].

#### 3.3.2. Core Independent Variables and Instrumental Variables

This study explored whether farmers’ participation in a cooperative reduces their use of chemical fertilizers and increases their use of organic fertilizers. Participation in cooperatives can positively impact farmers’ technology adoption behavior [64]. According to Ramirez and Ana [64], Abebaw and Haile [35], and Ma et al. [29], the indicator in this study is “whether or not a farmer joins a cooperative.” The farmers’ membership to a cooperative is anticipated to minimize the use of chemical fertilizers and boost the use of organic fertilizers. Since rural areas typically contain social networks [31,65], the presence of a cooperative in a village influences whether farmers join a cooperative, but is unrelated to farmers’ green technology adoption behavior. Therefore, the indicator of the presence of a cooperative in the village was chosen as an instrumental variable.

#### 3.3.3. Control Variables

This study specified individual, home, and company characteristics as the control variables. This study established five variables—age, gender, years of education, risk preference, and perception of green production—based on a review of the literature on individual characteristics [65,66,67]. Three variables—income status, social relationships, and loan amount—were defined based on peer studies on household characteristics [22,64,68,69]. Four variables—land acreage, stability of property rights, non-farm employment, and government support—were defined based on the literature on business characteristics [25,70]. Table 1 displays the descriptive statistics of the variables considered in this study.

## 4. Results and Discussion

### 4.1. Results

The regression results are summarized in Table 2. To avoid including farmers that use organic fertilizers, the estimate for farmers joining a cooperative based on the cost of chemical fertilizer was selected exclusively for the sample of farmers that did not use organic fertilizers.

First, the estimation results indicate that forming a cooperative can improve the likelihood of farmers reducing their chemical fertilizer usage, which is supported by earlier studies [71,72,73]. However, probably because of the heterogeneity of the crops studied, the findings contradict those of Ma et al. [10]. Second, the results show that membership of a cooperative increases the likelihood of using organic fertilizer, which is consistent with other major studies [10,74,75]. Farmers who join cooperatives are more likely to increase their use of soil-conserving fertilizer. In addition, farmers who join cooperatives are likely to reduce their fertilizer expenses without applying organic fertilizer. These findings are consistent with some studies [71,76,77], but not with the findings of Ma et al. [10], possible due to the different crops studied. In addition, these results contradict the findings of Abebaw and Haile [35] due to the low degree of economic development and limited purchasing power of farmers in Africa, who tend to apply too little fertilizer [78], which contrasts with China’s excessive use of chemical fertilizers. This study also employed an endogenous transformation model in conjunction with the endogenous transformation probit model for empirical estimation, which further confirmed the robustness of the estimation results, as illustrated in Table 3.

### 4.2. Treatment Effect Estimation

After obtaining the regression coefficients of the selection equation and the outcome equation, the individual treatment effects of the treatment group, the control group, and the entire sample were calculated separately based on the above models. The average treatment effects of the treatment group, the control group, and the entire sample were calculated by adding them separately and dividing them by their respective sample sizes. The results are summarized in Table 4. The average treatment effect on treatment (ATT) was 0.456 for the probability of reduced fertilizer application. This indicates that, for farmers who joined the cooperative, the probability of reducing fertilizer application was 35.6% higher than if they had not joined the cooperative. The average treatment effect (ATE) was 0.354, indicating that the probability of reducing fertilizer application increased by 35.4% if all farmers joined a cooperative. For farmers who had not joined the cooperative, the probability of reducing fertilizer application increased by 33.0% if they joined the cooperative. Their ATT was 0.220, indicating that the probability of them choosing to apply organic fertilizer would be reduced by 22.0% if they had not joined a cooperative. The average treatment effect on the control (ATC) was 0.834, indicating that for farmers who had not joined the cooperative, the probability of choosing organic fertilizer improved by 83.4%. The population of ATE was 0.718, indicating that if all farmers joined the cooperative, the probability of opting to apply organic fertilizer would increase by 71.8%. Therefore, joining a cooperative encourages smallholder farmers to utilize organic fertilizer.

### 4.3. Discussion

The study indicates that cooperatives can greatly encourage smallholder farmers to embrace green production practices; however, restrictions can hinder the ability of cooperatives to get smallholder farmers to embrace new technology. Further studies are therefore necessary.

First, land area scale is a key variable. After joining a cooperative, farmers with smaller acreages are more inclined to cut fertilizer application because smaller-scale farmers are typically less conversant with scientific application doses than their larger-scale counterparts. When they are not members of a cooperative, small-scale farmers tended to overuse fertilizer. After joining a cooperative, small-scale farmers increase their knowledge of scientific agricultural techniques and are thus more likely to limit their use of chemical fertilizers. In terms of organic fertilizer application, large-scale farmers who join cooperatives are able to significantly enhance their likelihood of application; however, the estimated outcomes for small-scale farmers are not significant. Large-scale farmers may be more specialized and more eager to increase their profits by applying organic fertilizers. Small-scale farmers are primarily part-time, not intensively involved in farming, and less interested in increasing farming earnings. Therefore, they are less willing to pay higher rates for organic fertilizers. By joining cooperatives, both large-scale and small-scale farmers can significantly lower the cost of fertilizer application, while the coefficient is greater for large-scale farmers. Due to the larger scale of their operations, large-scale farmers are more sensitive to changes in fertilizer costs and, as a result, do not apply excessive amounts of fertilizer, but rather attempt to cut fertilizer costs as much as possible. The regression results are summarized in Table 5.

Second, more attention needs to be paid to farmers’ perceptions of green production. In terms of fertilizer application reduction, joining a cooperative is insignificant for farmers with green knowledge. Farmers who are aware that chemical fertilizers are detrimental tend to voluntarily reduce their use of chemical fertilizers without a cooperative’s help. However, membership to a cooperative has a considerable impact on the reduction of fertilizer among smallholder farmers who previously lacked green knowledge. After receiving a cooperative’s guidance, farmers who were unaware of the adverse consequences of using chemical fertilizers reduce the intensity of chemical fertilizer application. Farmers with green knowledge who join a cooperative have a considerably higher probability of applying organic fertilizer. Green-conscious farmers tend to pay more attention to the quality of agricultural products. Therefore, they were more likely to pay a premium for organic fertilizer after joining a cooperative. After joining a cooperative, smallholder farmers who lack environmental awareness have no substantial impact on the application of organic fertilizer. Farmers who lack green expertise do not believe that it is important to apply organic fertilizer at a higher price; therefore, even if they join a cooperative, they will not use organic fertilizer. Both farmers who have green knowledge and those who do not could reduce their fertilizer costs by joining a cooperative; however, the coefficient is greater for the sample of green-aware farmers. Environmentally conscious farmers are therefore more inclined to reduce fertilizer use after joining a cooperative. As a result, farmers with a stronger concern for the environment who join the cooperative enjoy a greater reduction in fertilizer expenses. The regression results are summarized in Table 6.

## 5. Conclusions and Policy Suggestions

China has one of the most influential farmer communities in global food production. The questions whether and how cooperatives in the farming sector can encourage farmers to engage in environmentally responsible production practices have become key issues. This study uses national survey data to characterize green production practices by fertilizer reduction and develops a counterfactual research approach based on an econometric model to assess the influence of joining cooperatives on smallholder farmers’ fertilizer reduction.

This study demonstrates that joining a cooperative has a beneficial effect on the selection of green production strategies by smallholder farmers. Comparatively, joining a cooperative increases the likelihood of reducing fertilizer application by 35.6% and the likelihood of adopting organic fertilizer by 22.0%, while decreasing the cost of fertilizer application by $209.2/ha. In addition, the influence of smallholder farmers joining cooperatives on the likelihood of fertilizer application reduction varies according to the scale of their farming and their green perception. Therefore, the scientific contributions of this study mainly focus on the following two points. First, this study found that joining cooperatives can promote small farmers to use fewer chemical fertilizers and more organic fertilizers. Second, after joining the cooperative, it is not a separate green production behavior for small farmers to reduce the use of chemical fertilizers. But organic fertilizer plays an alternative role in chemical fertilizer.

There are some policy suggestions. First, cooperatives should be actively supported for serving farmers. To increase their market competitiveness, cooperatives provide smallholder farmers with more cost-effective agricultural material and technical services. Second, the policy system of cooperatives should be improved. Financial and taxation support should focus on cooperatives that primarily unite smallholder farmers to guide the healthy development of various types of cooperatives. Third, we should enhance small farmers’ understanding of cooperatives to better promote the green transformation of small farmers’ production behavior. Fourth, we should support cooperatives to develop and cultivate large-scale farmers. Cooperatives should be encouraged to cultivate large-scale farmers to achieve the popularization and promotion of new agricultural technologies.

We acknowledge some limitations of our study. Although we examined the application cost of chemical fertilizers, we did not precisely measure their dangerous component content. Therefore, further research is required to determine the effects of adding chemical fertilizer to soil and crops. Increasing the usage of organic fertilizers and decreasing the usage of chemical fertilizers is simply one part of environmental friendliness. Reducing the usage of pesticides and using straw-returning technologies are other environmentally friendly farming practices. However, this study focuses mostly on chemical and organic fertilizers. Therefore, future studies should emphasize the adoption of eco-friendly production techniques.

## Figures and Tables

**Figure 1 ijerph-19-16647-f001:**
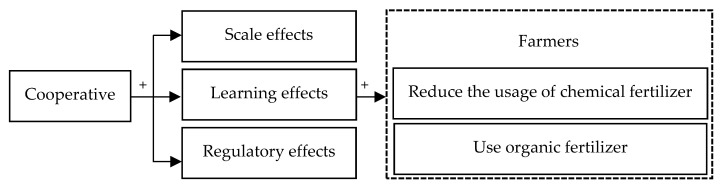
Theoretical framework.

**Figure 2 ijerph-19-16647-f002:**
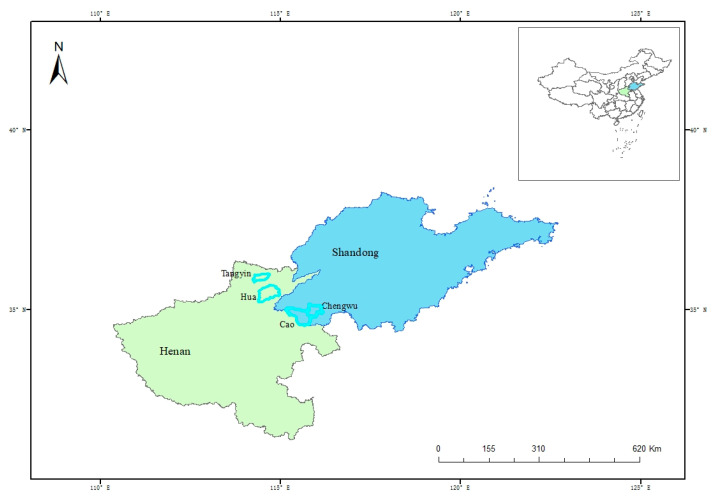
Maps of the study sites.

**Table 1 ijerph-19-16647-t001:** Variable definitions and summary statistics.

Category	Variables	Variables’ Meaning	Unit	Mean	Std.	Min	Max
Key dependent variables	NPK	Whether or not fertilizer is applied in reduced quantities	Yes = 1; No = 0	0.188	0.391	0	1
	organic	Whether to use organic fertilizer	Yes = 1; No = 0	0.212	0.409	0	1
	cost	Cost of fertilizer per hectare	$/ha	1028.866	135.49	434.95	1300.49
Key independent variable	coop	Whether to join a cooperative	Yes = 1; No = 0	0.188	0.391	0	1
Instrumental variable	coop1	Whether there is a cooperative in the village	Yes = 1; No = 0	0.219	0.414	0	1
Individual characteristics	age	Actual age in interview year	Age	49.802	8.914	31	70
	gender	Gender of the farmer	Men = 1; women = 0	0.796	0.403	0	1
	edu	How many years of schooling in total	Year	6.535	2.534	0	13
	risk	The type of risk preference	Preferred risk = 5; generally preferred risk = 4; risk neutral = 3; generally risk averse = 2; risk averse = 1	2.412	1.1	1	5
	cognition	Whether the farmer knows that chemical fertilizer is harmful to the quality and safety of agricultural products	Yes = 1; No = 0	0.249	0.433	0	1
Family characteristics	income	Annual household income per capita	$	766.96	414.65	434.95	3044.63
	relation	Whether there are any civil servants among the farmer’s family and friends	Yes = 1; No = 0	0.274	0.446	0	1
	loan	Current loan amount	$	21752	12700	0	43495
Business characteristics	land	Actual area of land in use	hectare	1.184	1.189	0.067	5
	rights	The proportion of land farmed with confirmed rights	%	89.826	9.201	70	100
	worker	The percentage of family members who are employed outside the home	%	33.895	17.211	0	80
	subsidy	Whether or not to receive government support policies	Yes = 1; No = 0	0.267	0.443	0	1

**Table 2 ijerph-19-16647-t002:** Baseline regression results for whether joining a cooperative reduces fertilizer application by smallholder farmers.

	NPK		Organic		Cost
	Probit	Marginal Effects	Probit	Marginal Effects	OLS
coop	1.203 ***	0.081 ***	0.868 **	0.068 ***	−0.962 ***
	(0.290)	(0.018)	(0.279)	(0.022)	(0.134)
age	−0.035	−0.002 *	−0.001	−0.000	−0.004
	(0.021)	(0.001)	(0.018)	(0.001)	(0.003)
gender	−0.941 ***	−0.063 ***	−1.022 ***	−0.080 ***	0.049
	(0.284)	(0.019)	(0.259)	(0.020)	(0.094)
edu	0.056	0.004	−0.028	−0.002	−0.001
	(0.070)	(0.005)	(0.065)	(0.005)	(0.011)
risk	0.139	0.009	0.060	0.005	−0.011
	(0.115)	(0.008)	(0.105)	(0.008)	(0.019)
cognition	0.169	0.011	0.336	0.026	0.002
	(0.328)	(0.022)	(0.307)	(0.024)	(0.053)
income	−0.635	−0.043 *	0.369	0.028	0.080
	(0.369)	(0.025)	(0.342)	(0.026)	(0.098)
relation	0.033	0.002	0.513	0.040 *	−0.020
	(0.372)	(0.025)	(0.299)	(0.023)	(0.054)
loan	0.028	0.002	−0.019	−0.002	−0.002
	(0.023)	0.002	(0.021)	(0.002)	(0.003)
land	0.032 ***	0.002 ***	0.029 ***	0.002 ***	−0.003 *
	(0.009)	(0.001)	(0.009)	(0.001)	(0.002)
rights	0.043	0.003	0.006	0.000	0.005
	(0.030)	(0.002)	(0.024)	(0.002)	(0.003)
worker	0.019 *	0.001 **	0.012 *	0.001 **	−0.002
	(0.007)	(0.000)	(0.006)	(0.000)	(0.001)
subsidy	1.029 **	0.069 ***	0.939 ***	0.074	−0.060
	(0.332)	(0.022)	(0.272)	(0.021)	(0.052)
_cons	1.846	-	−2.492	-	4.792 ***
	(3.028)	—	(2.509)	-	(0.319)
N	712	712	712	712	561
r2	-	-	-	-	0.254

Note: standard errors are in parentheses; * *p* < 0.05, ** *p* < 0.01, *** *p* < 0.001.

**Table 3 ijerph-19-16647-t003:** Estimated results of joining cooperatives on the reduction of fertilizer application by smallholder farmers (endogenous transformation model).

	Selected Equ.	Resulting Equ.		Selected Equ.	Resulting Equ.		Selected Equ.	Resulting Equ.	
	Coop	NPK0	NPK1	Coop	Organic0	Organic1	Coop	lncost0	lncost1
age	−0.015	−0.002	−0.003	−0.012	−0.001	0.012	−0.028	−0.001	0.001
	(0.026)	(0.001)	(0.006)	(0.028)	(0.001)	(0.006)	(0.027)	(0.001)	(0.003)
gender	−1.497 ***	−0.253 ***	0.002	−1.796 ***	−0.388 ***	0.046	−1.802 ***	0.039 *	−0.016
	(0.387)	(0.030)	(0.081)	(0.387)	(0.028)	(0.078)	(0.404)	(0.016)	(0.030)
edu	0.165	−0.002	0.021	0.212 *	0.003	0.001	0.171	0.005 *	0.002
	(0.087)	(0.005)	(0.019)	(0.096)	(0.004)	(0.019)	(0.092)	(0.002)	(0.007)
risk	0.055	0.002	0.023	0.124	0.029 ***	−0.046	0.104	0.004	−0.003
	(0.159)	(0.008)	(0.028)	(0.166)	(0.007)	(0.028)	(0.172)	(0.004)	(0.011)
cognition	0.526	0.008	0.156	0.360	0.066 **	0.075	0.538	0.017	0.002
	(0.385)	(0.022)	(0.089)	(0.447)	(0.020)	(0.090)	(0.418)	(0.011)	(0.036)
income	−0.251	−0.021	−0.086	−0.341	0.223 ***	−0.169	−0.391	0.027	−0.041
	(0.451)	(0.038)	(0.096)	(0.484)	(0.035)	(0.096)	(0.476)	(0.019)	(0.038)
relation	−1.303 *	0.029	0.051	−1.713 *	0.052 *	0.043	−1.641 *	0.011	−0.021
	(0.604)	(0.024)	(0.103)	(0.678)	(0.022)	(0.102)	(0.646)	(0.012)	(0.041)
loan	0.034	−0.004 **	0.012 *	0.021	−0.005 ***	0.008	0.012	0.000	0.001
	(0.028)	(0.001)	(0.006)	(0.028)	(0.001)	(0.006)	(0.029)	(0.001)	(0.002)
land	0.016	0.003 ***	0.001	0.023	0.003 ***	0.009 **	0.021	−0.002 ***	−0.004 ***
	(0.011)	(0.001)	(0.003)	(0.013)	(0.001)	(0.003)	(0.012)	(0.000)	(0.001)
rights	−0.032	−0.002	0.018	−0.005	−0.000	−0.009	−0.013	0.002 **	−0.000
	(0.040)	(0.001)	(0.012)	(0.045)	(0.001)	(0.012)	(0.045)	(0.001)	(0.005)
worker	−0.020 *	0.003 ***	0.000	−0.024 **	0.003 ***	−0.004 *	−0.026 **	−0.002 ***	−0.002 *
	(0.009)	(0.001)	(0.002)	(0.009)	(0.001)	(0.002)	(0.009)	(0.000)	(0.001)
subsidy	0.312	0.087 ***	0.164	0.386	0.124 ***	0.007	0.458	0.003	−0.046
	(0.396)	(0.023)	(0.101)	(0.440)	(0.021)	(0.103)	(0.436)	(0.011)	(0.041)
coop1	3.129 ***	-	-	3.159 ***	-	-	3.130 ***	-	-
	(0.429)	-	-	(0.481)	-	-	(0.460)	-	-
_cons	0.821	0.474 ***	−1.368	−1.740	0.218	1.124	0.422	6.074 ***	6.217 ***
	(3.853)	(0.139)	(1.172)	(4.480)	(0.127)	(1.205)	(4.208)	(0.071)	(0.481)

Note: standard errors are in parentheses; * *p* < 0.05, ** *p* < 0.01, *** *p* < 0.001.

**Table 4 ijerph-19-16647-t004:** Treatment effects of joining cooperatives on reduced fertilizer application by smallholder farmers.

	Endogenous Transformation Probit Model/Endogenous Transformation Model		
	ATT	ATU	ATE
Probability of fertilizer application reduction	0.456 ***	0.330 ***	0.354 ***
	(0.239)	(0.299)	(0.286)
Probability of organic fertilizer application	0.220 ***	0.834 ***	0.718 ***
	(0.296)	(0.347)	(0.412)

Note: standard errors are in parentheses; *** *p* < 0.001.

**Table 5 ijerph-19-16647-t005:** Scale heterogeneity analysis of joining cooperatives on smallholder farmers to reduce fertilizer application.

	NPK		Organic		Cost	
	High-Scale	Low-Scale	High-Scale	Low-Scale	High-Scale	Low-Scale
coop	1.272 **	1.903 **	1.115 *	0.561	−1.668 ***	−0.699 ***
	(0.483)	(0.595)	(0.509)	(0.428)	(0.454)	(0.151)
Control variables	Controlled	Controlled	Controlled	Controlled	Controlled	Controlled
N	241	471	241	471	121	440

Note: standard errors are in parentheses; * *p* < 0.05, ** *p* < 0.01, *** *p* < 0.001.

**Table 6 ijerph-19-16647-t006:** Heterogeneity analysis of green perceptions of smallholder farmers’ fertilizer application reduction by joining cooperatives.

	NPK		Organic		Cost	
	Cognition = 1	Cognition = 0	Cognition = 1	Cognition = 0	Cognition = 1	Cognition = 0
coop	1.546	2.208 ***	2.244 *	0.595	−1.021 *	−0.888 ***
	(0.915)	(0.588)	(0.880)	(0.367)	(0.411)	(0.154)
Control variables	Controlled	Controlled	Controlled	Controlled	Controlled	Controlled
N	177	535	177	535	79	482

Note: standard errors are in parentheses; * *p* < 0.05, *** *p* < 0.001.

## Data Availability

If necessary, we can provide raw data.

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
