# Peer review of "Will Joining Cooperative Promote Farmers to Replace Chemical Fertilizers with Organic Fertilizers?"

_ijerph, 2022, doi:10.3390/ijerph192416647_

Round 1

Reviewer 1 Report

Chemical fertilizer has been one of the sources of agricultural pollution. Examining farmers’ decision behavior in adopting organic fertilize is of significance for the sustainable agricultural development. This manuscript employs an endogenous conversion model to explore the impact of joining cooperatives on the reduction of chemical fertilizer consumption and organic fertilizers in China with field investigation in four counties in Henan and Shangdong. Overall, I think the topic of this study is interesting and the method used in this study makes sense. Still, some points need to be reconsidered.

1.  The title can be more specific as the impact of “cooperative” includes a lot of things. It can be replaced with “Will joining cooperatives promote farmers to replace chemical fertilizers with organic fertilizers?”

2.  The research gap should be added in Introduction. The motivation can be improved by introducing more international studies. In addition, how cooperatives impact small-scale farmers’ production behavior should be discussed more sufficiently.

3.           The theoretical analysis can be improved by discussing more about how scale effect, learning effect and regulatory effect play roles in reducing chemical fertilizer and increasing organic fertilizer.

4.           The reasons why choosing three towns and what the chosen towns are should be clarified in Section 3.1. Additionally, to be more rigorous, the affiliation of research team (line 143) can be removed from the text.

5.           The title of Section 4.1 (Figures, Tables and Schemes) can be changed. For example, it can be replaced with “4.1 Results”.

6.           The discussion can be improved by comparing with the relevant research at home and abroad.

7.           It would be better if the scientific contribution and novel knowledge of this research can be more specified in Conclusion.

8.           I recommend the authors double check the language after revision. There are some grammatical mistakes. And the tense of verbs in this article should be consistent.

Author Response

Response to Reviewer 1 Comments

Thank you very much for your comments and professional advice. These opinions help to improve academic rigor of our article. Based on your suggestion and request, we have made corrected modifications on the revised manuscript. We hope that our work can be improved again.

Point 1: The title can be more specific as the impact of “cooperative” includes a lot of things. It can be replaced with “Will joining cooperatives promote farmers to replace chemical fertilizers with organic fertilizers?”

Response 1: Thank you for your suggestion and the title has been replaced.

Point 2: The research gap should be added in Introduction. The motivation can be improved by introducing more international studies. In addition, how cooperatives impact small-scale farmers’ production behavior should be discussed more sufficiently.

Response 2: We have supplemented the blank part of the research in the introduction of this paper.

We also supplemented more studies to the three effects to support the viewpoint of this study. Based on the analysis of the existing literature, this study summarized the influence paths of cooperatives on reducing the use of chemical fertilizers and increasing the use of organic fertilizers by farmers into three key effects, such as scale, learning and supervision. There may exist other influence paths. However, limited by our time and funding budget, this study only analyzes the most important three paths. The added references information as follows:

[1] Savari, M.; Yazdanpanah, M.; Rouzaneh, D. Factors affecting the implementation of soil conservation practices among Iranian farmers. Sci. Rep., 2022, 12(1), 1-13.

[2] Savari, M.; Damaneh, H. E.; Damaneh, H. E. Factors involved in the degradation of mangrove forests in Iran: A mixed study for the management of this ecosystem. J. Nat. Conserv., 2022, 66, 126153.

[3] Zhu, P.; Zheng, J.; Zhang, M.; Zhao, X. Can participating in cooperatives promote the adoption of green production technologies by food and agriculture? —— From the perspective of endogenous power and external constraints. World Agri., 2022, 11, 71-82. [in Chinese]

[4] Wan, L.; Cai, H. Research on the influence of cooperative participation on the adoption of formula fertilization technology by farmers-based on the perspective of standardized production, Agri. Tech. Econ., 2021, 3, 63-77. [in Chinese]

[5] Feng, X.; Huo, X. Research on the social network's incentive for farmers to adopt environmentally friendly technologies. J. Chongqing Univ. 2016, 22, 72-81. [in Chinese]

[6] Uhunamure, S.E.; Kom, Z.; Shale, K.; Nethengwe, N.S.; Steyn, J. Perceptions of Smallholder Farmers towards Organic Farming in South Africa. Agriculture 2021, 11, 1157, https://doi.org/10.3390/agriculture11111157

[7] Kilic, N.; Burgut, A.; Gündesli, M.A.; Nogay, G.; Ercisli, S.; Kafkas, N.E.; Ekiert, H.; Elansary, H.O.; Szopa, A. The Effect of Organic, Inorganic Fertilizers and Their Combinations on Fruit Quality Parameters in Strawberry. Horticulturae 2021, 7, 354, https://doi.org/10.3390/agriculture11111157

[8] Fang, P.; Abler, D.; Lin, G.; Sher, A.; Quan, Q. Substituting Organic Fertilizer for Chemical Fertilizer: Evidence from Apple Growers in China. Land 2021, 10, 858. https://doi.org/10.3390/land10080858

[9] Sapbamrer, R.; Thammachai, A. A Systematic Review of Factors Influencing Farmers’ Adoption of Organic Farming. Sustainability 2021, 13, 3842. https://doi.org/10.3390/su13073842

[10] Sradnick, A.; Feller, C. A Typological Concept to Predict the Nitrogen Release from Organic Fertilizers in Farming Systems. Agronomy 2020, 10, 1448. https://doi.org/10.3390/agronomy10091448

Point 3: The theoretical analysis can be improved by discussing more about how scale effect, learning effect and regulatory effect play roles in reducing chemical fertilizer and increasing organic fertilizer.

Response 3: We have added some new references to the three effects and improved the theoretical analysis to support the viewpoints of this study. The added references information as follows:

[1] Zhu, P.; Zheng, J.; Zhang, M.; Zhao, X. Can participating in cooperatives promote the adoption of green production technologies by food and agriculture? From the perspective of endogenous power and external constraints. World Agri., 2022, 11, 71-82. [in Chinese]

[2] Wan, L.; Cai, H. Research on the influence of cooperative participation on the adoption of formula fertilization technology by farmers-based on the perspective of standardized production, Agri. Tech. Econ., 2021, 3, 63-77. [in Chinese]

[3] Feng, X.; Huo, X. Research on the social network's incentive for farmers to adopt environmentally friendly technologies. J. Chongqing Univ. 2016, 22, 72-81. [in Chinese]

Point 4: The reasons why choosing three towns and what the chosen towns are should be clarified in Section 3.1. Additionally, to be more rigorous, the affiliation of research team (line 143) can be removed from the text.

Response 4: We have deleted the corresponding part in the article. In addition, we added the reasons for choosing three towns and added the names of the selected towns.

Point 5: The title of Section 4.1 (Figures, Tables and Schemes) can be changed. For example, it can be replaced with “4.1 Results”.

Response 5: Thank you for your suggestion and it has been revised.

Point 6: The discussion can be improved by comparing with the relevant research at home and abroad.

Response 6: In our study, there are domestic and foreign literature for comparison. Of course, we have also added some new literature. For example, Zhu et al. (2022), Wan and Cai (2021), Feng and Hu (2016), Xu and Wu(2018), Yuan (2013) and Huang (2008). As follows:

[1] Zhu, P.; Zheng, J.; Zhang, M.; Zhao, X. Can participating in cooperatives promote the adoption of green production technologies by food and agriculture? —— From the perspective of endogenous power and external constraints. World Agri., 2022, 11, 71-82. [in Chinese]

[2] Wan, L.; Cai, H. Research on the influence of cooperative participation on the adoption of formula fertilization technology by farmers-based on the perspective of standardized production, Agri. Tech. Econ., 2021, 3, 63-77. [in Chinese]

[3] Feng, X.; Huo, X. Research on the social network's incentive for farmers to adopt environmentally friendly technologies. J. Chongqing Univ. 2016, 22, 72-81. [in Chinese]

[4] Xu, X.; Wu, B. Are cooperatives an ideal carrier for the organic connection between small farmers and modern agricultural development?. China Rural Econ., 2018, 11, 80-95.

[5] Yuan, P. "Company+Cooperative+Farmers" under the four modes of agricultural industrialization from the perspective of improving farmers' welfare. China Rural Econ., 2013, 4, 71-78.

[6] Huang, Z. Some theoretical and practical problems in the development of farmers' cooperative organizations in China. China Rural Econ., 2008,11,4-7.

Point 7: It would be better if the scientific contribution and novel knowledge of this research can be more specified in Conclusion.

Response 7: We have added the supplements to our scientific contributions in conclusion section.

Point 8: I recommend the authors double check the language after revision. There are some grammatical mistakes. And the tense of verbs in this article should be consistent.

Response 8: We agree with this suggestion and have double checked the grammar and the terminology throughout the text as appropriate. Actually, we used the Editage’s language polishing service for our first submitted version. If necessary, we can contact Editage to provide the proof of language polishing.

Reviewer 2 Report

This article is well organized. But for printing, it needs some improvements:

- It is better to provide more information in the abstract section

- In the materials and methods section, talk more about how to collect information as well as the study area.

- Presenting practical policies can help the attractiveness of this study

Good luck

Author Response

Response to Reviewer 2 Comments

Thank you very much for your comments and professional advice. These opinions help to improve academic rigor of our article. Based on your suggestion and request, we have made corrected modifications on the revised manuscript. We hope that our work can be improved again.

Point 1: It is better to provide more information in the abstract section

Response 1: Thank you for your suggestion and the data resources information has been added in the abstract section.

Point 2: It is better to use the following two articles in the introduction.

Response 2: Thank you for your recommendation. We have already supplemented these two references. They can be seen in references list [2] and [4].

Point 3: In the materials and methods section, talk more about how to collect information as well as the study area.

Response 3: In the part of data collection, we supplemented and improved the details of process of recording the questionnaire.

“Specifically, we chose Pulianji, Qingguji and Taoyuanji in Cao County; Wenshangji, Baifutu and Datianji in Chengwu County; Han Zhuang, Baiying and Guxian in Tangyin county; Daokou, Baidaokou and Liugu in Hua county. “

“The questionnaire was processed through one-to-one interviews between well-trained investigators and anonymous interviewees. Interviewees filled out questionnaires according to the facts. The interview places were all arranged in local closed offices that were conducive to the interviewees speaking freely. ”

Point 4: Presenting practical policies can help the attractiveness of this study.

Response 4: Thank you for your suggestion. The policy suggestions of our article are supplemented .

“There are some policy suggestions. First, cooperatives should be actively supported for serving farmers. To increase their market competitiveness, cooperatives provide smallholder farmers with more cost-effective agricultural material and technical services. Second, the policy system of cooperatives should be improved. Financial and taxation support should focus on cooperatives that primarily unite smallholder farmers to guide the healthy development of various types of cooperatives. Third, we should enhance small farmers' understanding of cooperatives to be better promoted the green transformation of small farmers' production behavior. Fourth, support cooperatives to develop and cultivate large-scale farmers. Cooperatives should be encouraged to cultivate large-scale farmers to achieve the popularization and promotion of new agricultural technologies.”

Reviewer 3 Report

This is a solid paper based on an original survey data set. The empirical analysis is rigorously done, the findings are clear, and the author's interpretation of the findings is sensible. There is, however, one major problem that must be addressed before the manuscript can be published: what is a cooperative in rural China? The authors neglected a key reality in rural China, that is: most cooperatives are not real cooperatives, but private agribusinesses simply registered as cooperatives. This has been repeatedly confirmed in the literature on farmers' cooperatives in China; the authors can refer to Hu et al.'s 2017 China Journal paper for a comprehensive account. 

The authors' review of the literature on cooperatives mostly draws on studies outside China [23, 25-29, 33-34], while the three references that studied China [30-32] all provided a negative evaluation of the operation and impact of cooperatives in rural China, which echoed my point above. 

What this means that the 2nd section on 'theoretical mechanisms' needs to be revised. There is no point discussing how theoretically cooperatives can change smallholders' farming practices when that is not how the so-called 'cooperatives' in China work -- unless the authors can present convincing evidence that the 'cooperatives' surveyed in the study really function as cooperatives. 

The authors' finding of the changes in 'co-op' members fertilizer use is still meaningful, but needs to be interpreted differently. The key reason that smallholders would shift from chemical fertilizer to the more expensive organic fertilizer is that their produce can be sold for higher prices (as organic wheat, for example). And this is possible, I imagine, because the agribusiness (albeit registered as cooperatives) that buy from them (either through contract farming or some other arrangements) are selling organic products. 

So, the real finding is that when farmers can either profitably shift to organic production (selling organic/green produce at higher prices) or gain more knowledge about green production, then they will reduce the use of chemical fertilizer. But there needs to be an agent that brings that change; this agent can be agribusinesses, private merchants, and the local governments, but can also be real cooperatives. It's just that real cooperatives are few and far between in rural China and the authors' data couldn't really tell whether those self-proclaimed cooperatives are real or not. 

Author Response

Response to Reviewer 3 Comments

Thank you very much for your comments and professional advice. These opinions help to improve academic rigor of our article. Based on your suggestion and request, we have made corrected modifications on the revised manuscript. We hope that our work can be improved again.

Point 1: This is a solid paper based on an original survey data set. The empirical analysis is rigorously done, the findings are clear, and the author's interpretation of the findings is sensible. There is, however, one major problem that must be addressed before the manuscript can be published: what is a cooperative in rural China? The authors neglected a key reality in rural China, that is: most cooperatives are not real cooperatives, but private agribusinesses simply registered as cooperatives. This has been repeatedly confirmed in the literature on farmers' cooperatives in China; the authors can refer to Hu et al.'s 2017 China Journal paper for a comprehensive account.

The authors' review of the literature on cooperatives mostly draws on studies outside China [23, 25-29, 33-34], while the three references that studied China [30-32] all provided a negative evaluation of the operation and impact of cooperatives in rural China, which echoed my point above.

What this means that the 2nd section on 'theoretical mechanisms' needs to be revised. There is no point discussing how theoretically cooperatives can change smallholders' farming practices when that is not how the so-called 'cooperatives' in China work -- unless the authors can present convincing evidence that the 'cooperatives' surveyed in the study really function as cooperatives.

The authors' finding of the changes in 'co-op' members fertilizer use is still meaningful, but needs to be interpreted differently. The key reason that smallholders would shift from chemical fertilizer to the more expensive organic fertilizer is that their produce can be sold for higher prices (as organic wheat, for example). And this is possible, I imagine, because the agribusiness (albeit registered as cooperatives) that buy from them (either through contract farming or some other arrangements) are selling organic products.

So, the real finding is that when farmers can either profitably shift to organic production (selling organic/green produce at higher prices) or gain more knowledge about green production, then they will reduce the use of chemical fertilizer. But there needs to be an agent that brings that change; this agent can be agribusinesses, private merchants, and the local governments, but can also be real cooperatives. It's just that real cooperatives are few and far between in rural China and the authors' data couldn't really tell whether those self-proclaimed cooperatives are real or not.

Response 1: Thank you for your suggestions but we politely disagree with the effective function of cooperatives. Although the references [30-32] in this paper do mention that cooperatives have many problems, including a weak driving capacity for farmers. However, there are also many high-quality domestic papers that indicated cooperatives still have strong driving capacity. In addition, we paid attention to strict sampling and control when we collected data.

The problem of weak driving capacity and the "empty shell" of some Chinese cooperatives does objectively exist, but this conclusion still requires caution in extending to our whole country. In fact, many well-known scholars in China have conducted case studies on a large number of cooperatives in China and demonstrated that cooperatives can benefit farmers, as in the following references [1-3]. In fact, we were aware of this issue during our data collection. Therefore, when we trained our surveyors, we explicitly required that only farmers who had participated in some training provided by cooperatives or had transactions with cooperatives were regarded as "joined cooperatives". Therefore, we believe that the data we obtained are true and valid. During the revision process, we added high-quality domestic references on the ability of cooperatives to motivate farmers, as follows.

[1] Xu, X.; Wu, B. Are cooperatives an ideal carrier for the organic connection between small farmers and modern agricultural development?. China Rural Econ., 2018, 11, 80-95.

[2] Yuan, P. "Company+Cooperative+Farmers" under the four modes of agricultural industrialization from the perspective of improving farmers' welfare. China Rural Econ., 2013, 4, 71-78.

[3] Huang, Z. Some theoretical and practical problems in the development of farmers' cooperative organizations in China. China Rural Econ., 2008,11,4-7.
